# Genomic Features and Evolution of the Parapoxvirus during the Past Two Decades

**DOI:** 10.3390/pathogens9110888

**Published:** 2020-10-27

**Authors:** Xiaoting Yao, Ming Pang, Tianxing Wang, Xi Chen, Xidian Tang, Jianjun Chang, Dekun Chen, Wentao Ma

**Affiliations:** 1College of Veterinary Medicine, Northwest A&F University, Yangling 712100, China; yaoxiaoting@nwafu.edu.cn (X.Y.); pangming@nwafu.edu.cn (M.P.); 2017011072@nwafu.edu.cn (T.W.); 2017011069@nwafu.edu.cn (X.C.); tangxidian@nwafu.edu.cn (X.T.); 2College of Agriculture and Animal Husbandry, Qinghai University, Xining 810016, China; 2018060204@nwafu.edu.cn; 3State Key Laboratory of Plateau Ecology and Agriculture, Qinghai University, Xining 810016, China

**Keywords:** parapoxvirus, *Poxviridae*, nucleotide composition, selection pressure, evolution

## Abstract

Parapoxvirus (PPV) has been identified in some mammals and poses a great threat to both the livestock production and public health. However, the prevalence and evolution of this virus are still not fully understood. Here, we performed an in silico analysis to investigate the genomic features and evolution of PPVs. We noticed that although there were significant differences of GC contents between orf virus (ORFV) and other three species of PPVs, all PPVs showed almost identical nucleotide bias, that is GC richness. The structural analysis of PPV genomes showed the divergence of different PPV species, which may be due to the specific adaptation to their natural hosts. Additionally, we estimated the phylogenetic diversity of seven different genes of PPV. According to all available sequences, our results suggested that during 2010–2018, ORFV was the dominant virus species under the selective pressure of the optimal gene patterns. Furthermore, we found the substitution rates ranged from 3.56 × 10^−5^ to 4.21 × 10^−4^ in different PPV segments, and the PPV VIR gene evolved at the highest substitution rate. In these seven protein-coding regions, purifying selection was the major evolutionary pressure, while the GIF and VIR genes suffered the greatest positive selection pressure. These results may provide useful knowledge on the virus genetic evolution from a new perspective which could help to create prevention and control strategies.

## 1. Introduction

Parapoxvirus (PPV) is a chordopoxvirus that causes non-systemic skin lesions in wild and domestic animals. PPV belongs to the *Poxviridae* family and contains orf virus (ORFV), pseudocowpox virus (PCPV), bovine papular stomatitis virus (BPSV), and parapoxvirus of red deer in New Zealand (PVNZ) [1,2,3]. The PPV genome is approximately 134–139 kb in size with an extremely high GC content, which goes up to 65% [4,5,6]. The genetic structures of PPVs exhibit a general pattern form of a central conserved region, with essential genes in regard to position, spacing, and orientation [5]. The terminal variable regions encode genes with important roles during viral infection such as the regulation of host immune response to infection and the determination of host range [5,6].

PPVs distributes globally, and disturbs the ruminant flocks in many continents [7,8,9,10]. This virus is known to infect mammals, such as goats, sheep, cattle, camels, reindeer, and other ruminants [7,11]. Human can be infected by PPVs, when directly exposed to skin lesions from infected mammals. Human PPVs cases are reported in a lot of regions of the world, including Europe, South America, and South Africa [7,12,13,14,15,16,17]. Currently, many live PPVs vaccines are available to control the disease, but they may not eradicate this disease [18].

ORFV, the prototype species of PPVs can cause lesions in the lips and breasts of suckling and grazing animals with high morbidity and low mortality rates [6,19]. Ruminants can be infected with ORFV more than once, with a shorter period of time to recovery and less obvious pathological changes than the primary infection [20]. BPSV can infect cattle of all ages, however, clinical signs are usually observed in calves. Like ORFV, repeated infection of ruminants with BPSV commonly occurs, indicating that BPSV may not confer effective immune response [6]. Besides ORFV and BPSV, PCPV and PVNZ are both maintained in mammals and threaten the animal health [6,21,22,23,24,25].

Vaccines have been indicated to impose a powerful selection pressure on the viral evolution [26,27]. Viruses may acquire greater virulence through genetic variation and recombination, as a result of the rapid transmission in vaccinated populations. For instance, deletions in the terminal locations have been discovered for vaccinia virus (VV) [28,29,30,31,32,33,34], rabbitpox virus [35,36], cowpox virus [37], and monkeypox virus [38]. The transposition of duplicated sequences was also been noticed for VV [39], cowpox virus [40,41], rabbitpox virus [36], and monkeypox virus [42]. These punctuated antigenic changes led to virus escape from the host immunity that was caused by earlier infection or vaccination, thus allowing the virus to re-infect their hosts who were once immune to virus and forcing the reformulation of virus [21,26,28]. Viruses also mutate readily due to low accuracy when duplicating their genome, which may assist the viruses to evade host immune attack and evolve further. PPVs have large genome, nearly 134–139 kb in size, where many replication errors may occur. Together with the selection pressure of vaccines, PPVs are even more likely to mutate [43,44].

This work reported the population dynamics of all available PPVs regarding the evolutionary process and genetic diversity. On the basis of PPVs genomes, we performed phylogenetic analysis and genomic comparison to investigate the genomic features of viral complete sequences. From the dataset of PPVs genome sequences, the evolutionary rate was also considered by Bayesian methods. Additionally, we further explored the different genes that have played an important role in PPVs evolution by estimating the genetic diversity and the selection pressure in this study.

## 2. Results

### 2.1. G and C Nucleotides Were Highly Enriched in PPV Genomes

To analyze the basic characteristic of the virus genome, the nucleotide contents of PPV whole genomes were calculated. It was shown that the mean compositions (%) of G (32.16 ± 0.32) and C (32.12 ± 0.22) were significantly higher than A (17.95 ± 0.28) and T (17.77 ± 0.25) (Appendix A, Figure 1A, *t*-test, *p* < 0.01). The mean percentages of G3 (36.92 ± 1.02) and C3 (37.12 ± 1.33) were also significantly higher than A3 (20 ± 1.02) and U3 (19.94 ± 0.89), highlighting that G- and C- ended codons are more likely to occur in the PPV whole genome sequences (Appendix A, Figure 1B, *t*-test, *p* < 0.01). According to the species classification, it was important to mention that a similar trend of nucleotide composition was observed among different species, also suggesting higher G/C or G3/C3 contents in PPV genomes. However, there were significant differences in the nucleotide composition parameters trends among four PPV species, including BPSV, PCPV, ORFV, and PVNZ (Appendix A, Figure 1C–D, *t*-test, *p* < 0.01). As shown in Figure 2, we further compared the nucleotide composition among four different PPV species. Strikingly, there were highly unusual patterns of nucleotide contents, and were significantly different from the nucleotide composition patterns in the various PPV species.

### 2.2. Different Genome Structures Exist in Various PPVs

According to the phylogenetic relationship and genomic structure, the PPV genera belongs to *Poxviridae* and includes four species: ORFV, PCPV, BPSV, and PVNZ (Figure 3). The genome of PPV contains a large number of protein-coding regions which were encoding functional proteins, such as major envelope protein (B2L), chemokine binding protein (CBP), immunodominant envelope protein (F1L), GM-CSF/IL-2 inhibition factor (GIF), vascular endothelial growth factor (VEGF), viral interferon resistance protein (VIR), and late transcription factor (VLTF). Besides these major functional proteins, different PPVs encode some special accessory proteins, including Ankyrin protein (ANK), IL-10-like protein (vIL-10), and poxviral anaphase promoting complex regulator/ring H2 protein (PACR) (Figure 3). These results suggested that compared with other PPVs, VRGF genes existed in the left terminal region of BPSV genomes. It was also of note that PCPV genomes contained one extra VEGF gene at the left end and lacked the vIL-10 gene, while vIL-10 replaced VEGF in the left end of GQ329669.1/PCPV genome. Additionally, PPVs contained more than one ANK genes that may be involved in host range.

### 2.3. Population Dynamics of PPVs over the Past 20 Years

In order to explore the changes of PPV deposited sequences in the world during the past 20 years, 266 available complete sequences of PPV B2L genes were retrieved from GenBank in this study (Appendix A). All B2L isolates were clustered into separate clades based on their phylogenetic tree, suggesting that ORFV was the major species, which circulated mainly in China and India (Figure 4A and Appendix A). Consistent with the pattern observed in the phylogenetic tree, the results of the case count also indicated that most of ORFVs were isolated from China and India, and their prevalence increased sharply during 2013–2015 (Figure 4B). Since 2010, ORFV has been the predominant species in circulation throughout China and India.

### 2.4. ORFV of Genera PPV Became Predominant Species in Recent Years

To describe the phylogenetic evolution of the prevalent PPVs, the phylogenetic trees were constructed for the seven major structural segments. Phylogenetic trees were partitioned into groups with a 20th percentile cutoff [45]. The PPV strains were not clustered exactly consistently in each tree of seven segments (Figure 5). Within each prevalent clade, a large number of ORFVs further formed a main group with few PCPV, BPSV, and PVNZ (Figure 5), demonstrating they may share a common ancestor surviving a bottleneck event. Since 2010, the species diversity of PPVs was reduced with widespread outbreak in many countries (Figure 4 and Figure 5), which can be reasonably considered by the high sequence similarity of PPVs isolated in the specific time span across broad regions.

In order to confirm this hypothesis, we estimated the temporal dynamic of gene sequence similarities of PPVs by collecting all available PPV sequences during 2000–2018. Pairwise identity comparison suggested a decreased diversity in all these seven segments of PPVs, starting mainly in 2010–2018 (Figure 6). At nearly the same time point, the reduced sequence diversity of almost all gene segments suggested that the virus with great fitness was selected, acquired dominance, and led to a widespread outbreak throughout the world. As we know, this is the first discovery of such genetic bottleneck in the PPV genomes (Figure 6), despite the prevalence of PPV in the world for the past 20 years.

### 2.5. Evolutionary Rates and Selection Analysis of Different Gene Sequences in PPVs

To understand the additional forces affecting PPV evolution, we analyzed the evolutionary rates and the average selective pressure of PPV strains whose isolation date were known. According to the different protein-coding gene sequences, a Bayesian coalescent approach was performed to infer the substitution rates of different segments. Based on the best-fit model, the Bayesian coalescent approach was performed to calculate the mean evolutionary rates of the seven gene segments, ranging from 3.56 × 10^−5^ to 4.21 × 10^−4^ substitutions per site per year (Table 1). The VEGF gene evolved at the slowest rate among the structural segments, with a substitution rate of 3.56 × 10^−5^ and the highest probability density (HPD) between 4.31 × 10^−6^ to 9.61 × 10^−5^. The VIR gene evolved at the highest estimated mean rate of 4.21 × 10^−4^ (95% HPD: 1.28 × 10^−4^–7.32 × 10^−4^).

The selective pressure for each segment was performed to suppose a constant substitution rate of all sites and provided average values across all sites. The GIF and VIR genes were found to be more variable, while the VLTF-1 gene was more conserved. On the basis of the Bayesian analysis, likelihood ratio tests were performed to compare the neutral models and the positive selection models. To estimate the selective pressure on each protein-coding segment, we applied a model (M3) for the differences among non-synonymous and synonymous rates (dN/dS) of amino acid residues, and several neutral models (M0, M1, and M7) with proportions of selected codons (M2 and M8) (Table 2). The log likelihood values showed that positive selection models (M2, M3, and M8) were more fitted in each gene regions than neutral models (M0, M1 and M7). The nested comparisons were also performed between the neutral models and the positive models, such as M0 vs M3, M1 vs M2 and M7 vs M8, and determined the better fitness of positive models, indicating that except for the VLTF-1, positive selection pressure occurred at different gene segments within the genome during the PPVs evolution process (chi-square test, *p* < 0.05) (Table 2).

## 3. Discussion

PPV infections normally occur in animals and even in humans worldwide [7,12,13,14,15,16]. However, comprehensive studies on PPVs are still lacking. In the present study, available whole genome sequences and different structural protein-coding sequences were both utilized to investigate the structural and evolutionary genomics of PPVs. To identify the common features and differences of PPVs, it is important to analyze genomic patterns of all representative sequenced PPV genomes. To start with, the general nucleotide contents were calculated for the genomes of different PPV species have been shown in Appendix A. The results suggested that the PPV coding sequences were GC-rich, especially on the third location of synonymous codons. It has been known that viruses can take on a wider range of GC compositions than other organisms [46,47]. Even within the same family, viruses have similar replication or life-cycle strategies, but may show different GC frequencies [48]. Although high GC amplification could have fixed erroneous base calls in the sequencing in an unpatterned and uneven manner, which might influence sequences reported and downstream analyses, we retrieved enough sequences and got the same conclusion with prior researches, which all indicated that the biological relevance of G/C richness and frequency [6,49,50].

These PPVs, although related, could be divided into four subgroups according to the phylogenetic tree (Figure 3), which contained ORFV, PCPV, BPSV, and PVNZ, respectively. It demonstrated a divergence among species affiliations based on complete genome sequences of PPVs. To extend the genomic description of these four subgroups of PPVs, we added the characterization of PPV genomes. PPV with large genomes (134–139 kb) encodes nearly complete replisomes, which may englobe more than one virus [4,5]. It has been suggested that VEGF plays an important role in PPV pathogenesis related to the vascularization process and epidermal lesion proliferation [6,51]. Consistent with previous studies, we identified that VEGF was located in the left terminal region of BPSV contrasting the right terminal position of ORFV, PCPV, and PVNZ VEGFs [6,52], which indicated that the divergence of BPSV VEGF may have distinct functions or binding specificities compared with other VEGFs. Notably, PCPV was divergent from other PPVs with a deficiency of vIL-10 or left terminal vIL-10. As a pleiotrophic cytokine, vIL-10 can generate immunosimulatory/immunosuppressive effects on cells. Additionally, vIL-10 is an important anti-inflammatory cytokine to inhibit non-specific immunity and plays the particular role in macrophage and Th1 effector [53,54]. Overall, these divergent genomic features may reflect the specific adaptation to their natural hosts [6].

In addition to the genomic characteristics of PPVs, the phylogenetic diversity of the segmented genome of PPV may be a crucial contributor to understanding the temporal and spatial patterns of viral outbreaks [55,56]. As shown in this study, the proportion of ORFV strains sequenced and available indicates a sharp increase in the past 20 years. Our results suggested that PPVs evolved under mutation pressure and natural selection over 20 years of prevalence in animals, and the predominant species, ORFV, was selected recently. Since 2010, ORFV has rapidly increased in many countries, especially in China and India, and is now the predominant PPV species in farmed livestock [57,58,59,60,61,62]. It has been indicated that under selective pressure, ORFV eventually became the dominant species during 2010–2018. The reduced sequence diversity of almost all gene segments at nearly the same time point indicates that the virus with greatest fitness was selected and resulted in a widespread outbreak throughout China and India.

Furthermore, according to the supposition of rate constancy, the differences in viral isolation dates can provide us with knowledge about the substitution rate of molecular evolution. The previous study has suggested that the magnitude of evolutionary change may accumulate since the isolation date [63]. A Bayesian coalescent approach suggested the mean substitution rates were between 3.56 × 10^−5^ to 4.21 × 10^−4^ in different gene segments. Besides, the highest substitution rate of the PPV VIR gene reflected its variability in viral synthesis and replication process than other gene segments, which may be caused by the long-lasting interaction with the immune response.

When we analyzed the selection profiles of PPV protein-encoding genes, the overall rates of dN/dS were mostly less than 1, demonstrating that purifying selection was the major force to drive the evolution of PPV genes. However, there were clear differences in this selection profiles of various genes. Among these protein coding genes, GIF and VIR were under greater positive selection than other gene segments, which were likely interacting with proteins of the immune system [18,64,65]. The variability of genes was constrained by the positive selection during adaptation, which is important for virus replication [66]. Our results suggested that VLTF-1 may suffer from greater purifying selection in the evolutionary process of PPVs. Additionally, the angiogenic factor, VEGF which has a huge influence on embryonic development or tumor neovascularization by binding to its receptor tyrosine kinases is reasonably conserved during the viral evolution in host populations. These results suggested that PPV genes are evolving at the rapid rate under selection pressure, which may improve viral fitness in their infected hosts. Therefore, it is likely that the PPV genomes have been reshaped by the animal immune system, although genetic mutation or recombination contribute to the natural evolution of PPVs.

## 4. Materials and Methods

### 4.1. PPVs Sequences

PPVs complete genome sequences and nucleotide sequences of the B2L, CBP, F1L, GIF, VIR, VEGF, and VLTF-1 genes were downloaded from GenBank (www.ncbi.nlm.nih.gov/genbank/). To explore the emergent time of the earliest PPV isolate, the sequences with an unknown collection date were excluded. The host, collection date, and sampling location of each PPV isolates was determined either from GenBank, or were obtained from the related publications. This led to a dataset of 17 complete genome sequences and 776 gene sequences, including 266 B2L genes, 38 CBP genes, 83 F1L genes, 133 GIF genes, 81 VIR genes, 122 VEGF genes, and 51 VLTF-1 genes, respectively (Appendix A). Partial sequences from the aforementioned genes were obtained from the complete genomes available at GenBank. The resulting sequences were aligned using MUSCLE v3.7 (http://www.drive5.com/muscle/) [67].

### 4.2. Gene Structure and Nucleotide Composition

The gene structure of PPV was visualized by comparing their genomic nucleotide sequences using the R program “gggenes” project. Nucleotide compositional analysis of the PPVs complete genome sequences was measured using BioEdit v7.2.5 (https://bioedit.software.informer.com/7.2/) [68], including the frequencies of A, T, G, C, AT, and GC. The nucleotide occurrence frequencies of the third codon position (T3, G3, C3, and A3) of synonymous codons were measured by the CodonW v1.3 (http://sourceforge.net/projects/codonw).

### 4.3. Compositional Change of PPV Isolates over Time

During 1974–2018, 266 B2L genes of PPV strains were downloaded from GenBank and analyzed with the phylogenetic analysis by maximum likelihood program (PAML v4.9, http://abacus.gene.ucl.ac.uk/software/paml.html), with 1000 bootstrap replicates. An online tool, the Interactive Tree Of Life v2, was performed to design the tree [69]. The resulting isolates were composed of 4 species in PPVs population, including ORFV, PCPV, BPSV, and PVNZ. Isolates were assigned based on the PPVs species and the initial collection country. The variation number of PPV was calculated using statistical methods and the plot was drawn with the project ggplot2 in R program [70].

### 4.4. Evolution Substitution Rates

For all PPV genes, the molecular evolution rates were estimated by the MCMC program BEAST v2.4.8 (http://beast.community). This program gives a maximum-likelihood estimate of the evolution substitution rate, based on a model that considered a constant evolution rate (molecular clock). We also estimated the confidence intervals for all the parameters.

To obtain the best-fitting models for each viral gene, jModelTest v2.1.7 was performed by Akaike Information Criterion (AIC) (https://www.softpedia.com/get/Science-CAD/jModelTest.shtml) [71], which was estimated for each model tested according to the approach of Newton and Raftery [72]. Here, we used general time reversible model, a strict molecular clock, and a coalescent model with constant available population size, as performed in the BEAST package. Bayesian Markov chain Monte Carlo (MCMC) analysis was implemented with 10 million steps and was sampled every 10,000 steps with 10% burn-in. Tracer v1.6 was used to analyze the results of running BEAST. The effective sample size was greater than 200 for each parameter assessment in the MCMC analysis. Statistical uncertainty in the dataset was estimated in the 95% HPD values.

### 4.5. Genetic Diversity and Phylogenetic Analysis

The genetic diversity (π) of PPV was evaluated as average pairwise difference between viral sequences, using the Tamura–Nei nucleotide substitution models performed in MEGA 5 [73]. BioEdit was implemented to estimate the sequence identity for different gene segments [68], and the heatmap was plotted by R program “pheatmap”project. For each PPV gene, bayesian phylogenetic tree was inferred by Bayesian Evolutionary Analysis Sampling Trees Software (BEAST) [74]. For all analyses we performed a coalescent constant population model and the general-time reversible model of nucleotide substitution [75]. Based on the correlation between collection date and evolutionary distance for each gene, a strict molecular clock was selected, as evaluated by TempEst [76]. Here, MCMC analysis was run for 10 million steps, 10% of which was removed as burn-in and parameters sampled every 10,000 steps. The effective sample size for all the traces was greater than 200.

### 4.6. Selection Pressures in PPVs

Selection pressure was highlighted as the ratio between the average number of non-synonymous nucleotide substitutions (dN) and synonymous (dS) nucleotide substitutions per site (dN/dS) and was implemented in the CODEML program within the PAML software package. Besides, paired models of dN/dS were estimated among nucleotide sequences, containing M0 versus M3, M2a versus M1a, and M8 versus M7. We applied the likelihood ratio test to compare the neutral and positive selection models, and chi-square test to indicate the divergence between the compared models. We also calculated the dN and dS values respectively.

## 5. Conclusions

In conclusion, this study indicated that PPVs share several common features within different virus species. However, we noticed that GC contents were significantly different between ORFV and other PPV species. The genetic characteristics were also not similar among all PPVs, which may have resulted from the specific adaptation to their natural hosts. Furthermore, the evolutionary features of PPVs suggested that ORFV has been the predominant PPV species and increased rapidly in many countries since 2010. PPV isolates have been evolving at a sharp increased rate under the natural selection pressure. Combining genetic mutation of PPVs, positive selection may reshape the PPV genomes during the evolutionary process. The genetic analysis of PPVs could extend our knowledge of the mechanisms that promote virus evolution.

## Figures and Tables

**Figure 1 pathogens-09-00888-f001:**
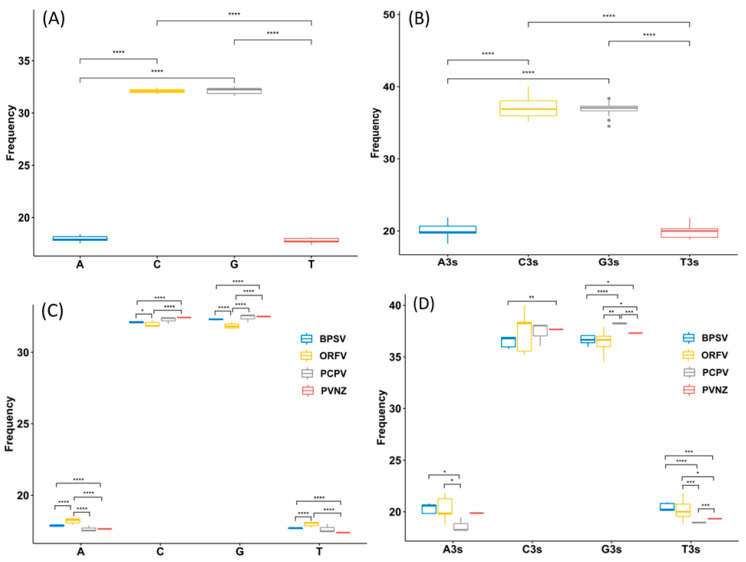
Nucleotide content distribution and composition. (**A**) The mean frequency for A, T, G, and C composition in 17 different parapoxvirus (PPV) sequences are shown, suggests higher GC content than AT on the whole. (**B**) The mean values of the nucleotide content frequency at the 3rd codon position. (**C**) Based on 4 various species of PPVs, analysis for the overall A, T, G, and C composition frequencies. (**D**) Based on 4 various species of PPVs, analysis for A, T, G, and C composition frequencies at the third codon position. Different asterisks represent statistical significance (* *p* < 0.05, ** *p* < 0.01, *** *p* < 0.001, **** *p* < 0.0001).

**Figure 2 pathogens-09-00888-f002:**
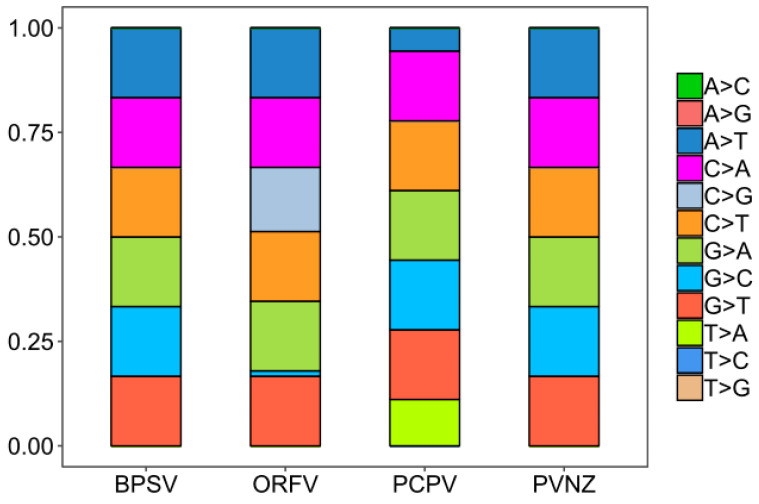
Nucleotide composition of PPVs for different species. The frequency of each observed nucleotide composition, reconstructed in BioEdit (v7.2.5), is shown for different species of PPVs.

**Figure 3 pathogens-09-00888-f003:**
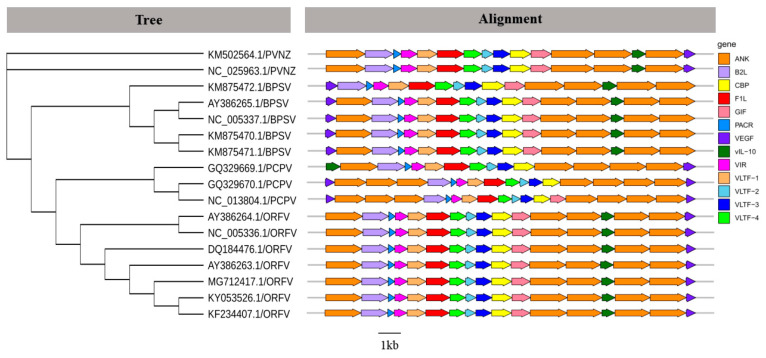
Phylogenetic analysis and genome comparison among different species of PPVs. According to Appendix A complete genome sequences of PPVs, Maximum Likelihood (ML) phylogenetic tree was reconstructed in phyML (v3.0) and tested with 1000 bootstraps. The predicted genes are represented by block arrows showing the direction of transcription. The genes encoding for structural proteins and replication-associated proteins of PPVs are shown in different colors.

**Figure 4 pathogens-09-00888-f004:**
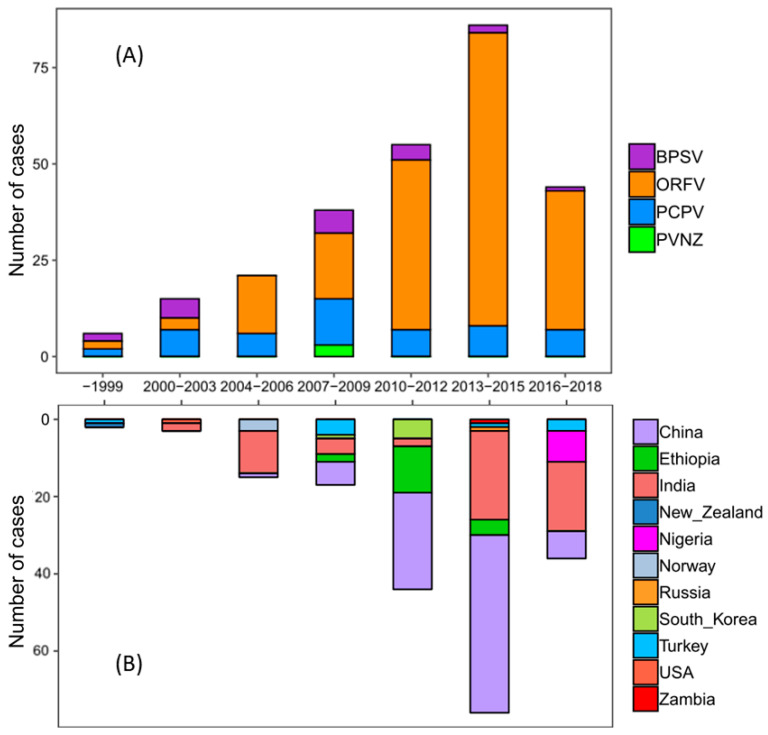
Population dynamics of PPVs during the past 20 years. Case counts are presented in the plot. The X-axis represents the year intervals, and the Y-axis represents the number of cases. Different colors represent different species (**A**) and different collection countries of PPVs (**B**).

**Figure 5 pathogens-09-00888-f005:**
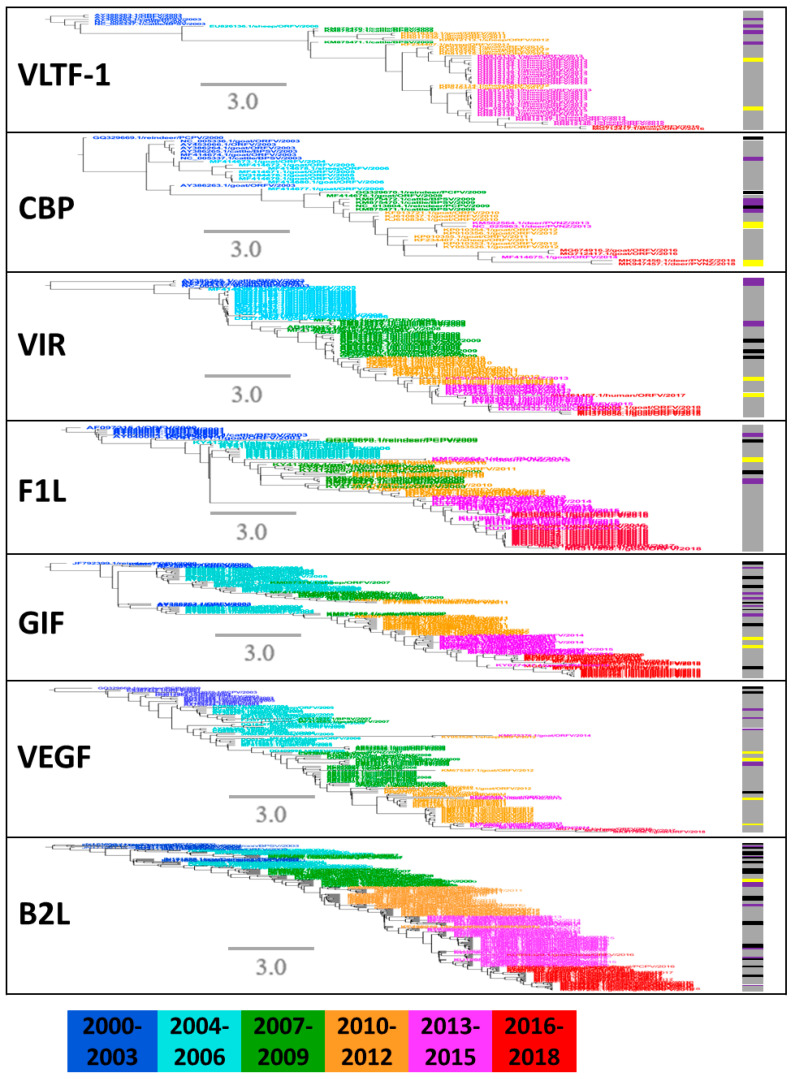
Phylogenetic analysis of seven structural genes with PPVs during 2000–2018. Clades with all of 2000–2018 viruses were fully shown. Color of line at right of each leaf node indicates year of isolation (see color bar). Timescale is in years. Vertical lines mark different species of PPVs: gray represents ORFV, purple represents BPSV, black represents PCPV and yellow represents PVNZ.

**Figure 6 pathogens-09-00888-f006:**
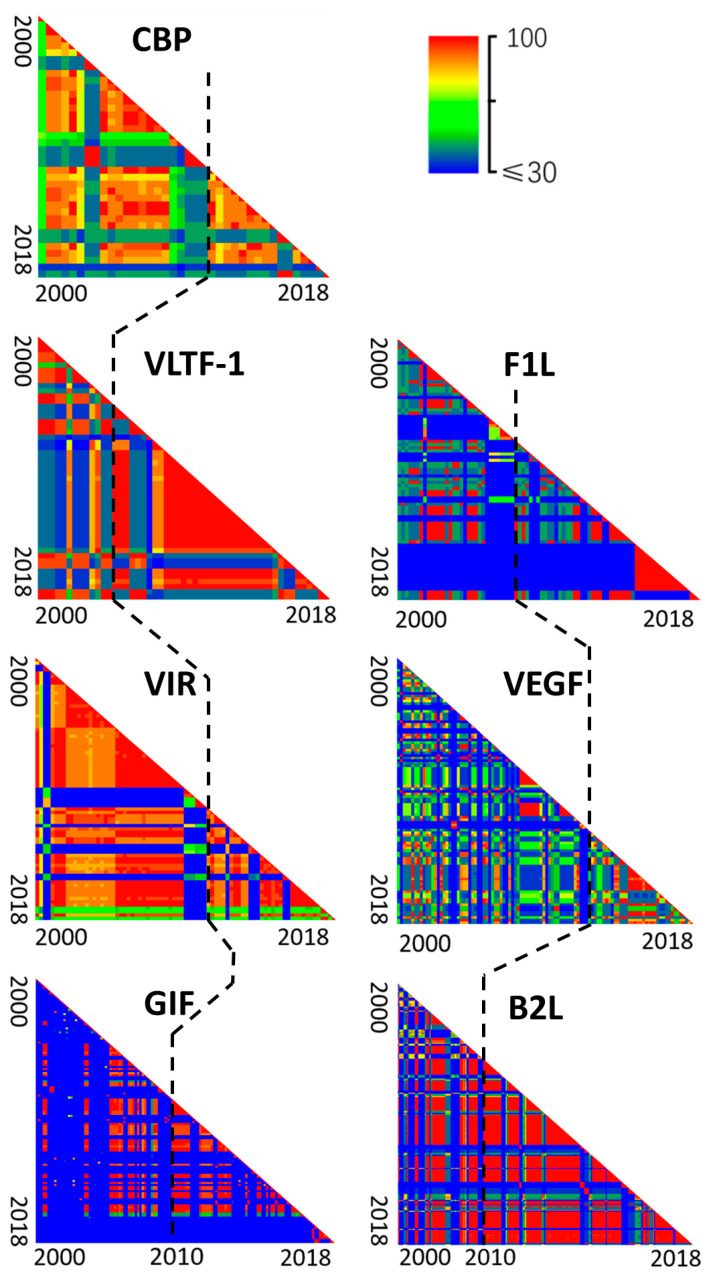
Decreased genetic diversity of the seven segments of PPVs during the widespread outbreaks in the world (2000–2018). Pairwise comparison of nucleotide sequences of PPVs was plotted as a heatmap. Viruses isolated from 2000 through 2018 were ordered by isolation date from left to right of the x axis and from top to bottom of the y axis. On the axes, the ticks indicated the isolation years. Black dotted line represented the year 2010. Color indicated identity levels from ≤30% (blue) to 100% (red).

**Table 1 pathogens-09-00888-t001:** Bayesian estimates of evolutionary rate of specific gene segments of *parapoxvirus*.

Gene	Evolutionary Rate (nt Substitutions per Site per Year)	95% HPD
B2L	6.34 × 10^−5^	3.89 × 10^−9^–1.52 × 10^−4^
CBP	3.51 × 10^−4^	2.39 × 10^−5^–7.06 × 10^−4^
F1L	2.79 × 10^−4^	1.39 × 10^−4^–4.31 × 10^−4^
GIF	1.29 × 10^−4^	3.53 × 10^−5^–3.07 × 10^−4^
VIR	4.21 × 10^−4^	1.28 × 10^−4^–7.32 × 10^−4^
VEGF	3.56 × 10^−5^	4.31 × 10^−6^–9.61 × 10^−5^
VLTF−1	4.45 × 10^−5^	1.85 × 10^−5^–7.22 × 10^−5^

HPD, highest probability density.

**Table 2 pathogens-09-00888-t002:** Likelihood ratio test was performed to compare positive selection models vs neutral models for different *parapoxvirus* gene segments.

Gene	M0 (ω)	Likelihood Test (2ΔlnL)		
		M0 vs. M3	M1a vs. M2a	M7 vs. M8
B2L	0.496	37.600 *	7.855 *	14.975 *
CBP	0.712	714.169 **	158.452 **	465.606 **
F1L	0.810	16.101 *	7.855 *	4.673 *
GIF	1.446	54.257 **	45.127 ***	45.116 ****
VIR	1.537	27.370 *	32.219 ***	39.494 ***
VEGF	0.801	64.918 ***	17.004 *	25.106 *
VLTF-1	0.015	1.062	1.766	0.001

Positive selection models: M2a, M3, M8; neutral models: M0, M1a, M7. Positive selection pressure was defined as dN/dS >1, where dN = number of non-synonymous substitutions per site and dS = number of synonymous substitutions per site. ω, dN/dS ratio; L, log likelihood; M0, one ratio model; M1, nearly neutral model; M2, positive selection model; M3, discrete model; M7, beta model; M8, beta and ω model. * *p* ≤ 0.05; ** *p* ≤ 0.01, *** *p* ≤ 0.001, **** *p* ≤ 0.0001.

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
