# Peer review of "Genomic Features and Evolution of the Parapoxvirus during the Past Two Decades"

_pathogens, 2020, doi:10.3390/pathogens9110888_

Round 1
Reviewer 1 Report
Yao et al. present here an interesting work in which phylogeny, genome structure and evolution features of Parapoxviruses using previously deposited sequences from 2000 to 2020 are depicted. Although the manuscript contains important and interesting knowledge, some important drawbacks impede its publication in the present form.
Major points:
1.- The manuscript needs deep English editing and usage revision. Some parts of the manuscript are really difficult to follow. Some examples as specific comments below.
2.- The manuscript lacks of accuracy and inconsistences in some parts. For example, PPV infection is claimed to cause high mortality rates.
3.- In general conclusions are overstated, i.e., the authors claim for an increased ORF prevalence without taking into account the availability of sequences. The possibility that countries suffering from Orf virus infection without reporting cases or uploading sequences to GenBank cannot be ruled out. Thus, conclusions should be softened. Parapoxvirus infections of ruminants and handlers are endemic worldwide. For example Lederman et al 2007. Clin Infect Dis. Taking into account that most sequences analyzed were from China and India, conclusions and objectives should be dimensioned.
4.- Phylogenetic evolution is based on 5 different genetic segments. Since the different genes exhibited different genetic features, such as different dN/ds ratio, why the authors think that evolution inferences would be that accurate? Complete genome analysis would offer a more precise picture.
Specific points:
Line 23: ORFV please define.
Line 24: This work is not unveiling mechanisms that determine the high GC content found in PPVs, since reasons could be variable, different mechanisms for the different viruses cannot be ruled out.
Line 28: Which outbreak the authors refer to? Please specify.
Lines 38-39: PPVs are also present in domestic ruminants. Please correct: “non-systemic skin lesions in wild and domestic animals”
Lines 47-48: The sentence is not clear. Please re-phrase.
Line 49: Better “Humans can be infected…”
Line 51: South America is repeated twice.
Line 54: ORFV infection is characterized by high morbility and low mortality rates. Please correct.
Lines 59-60: The last two sentences of the paragraph are confusing. Please rephrase.
Lines 62-67: The whole paragraph is awkward and confusing. No relation with vaccination is presented in the manuscript, and the selection pressure due to vaccination is not clearly explained.
Line 68: This seems to be overstated. Since PPVs are endemic in USA or Mediterranean countries such as Italy or Spain, and no data are analyzed in this work, I would suggest to soften some global conclusions.
Line 70: Investigating genomic features of different strains would imply isolation of the corresponding viruses and genetic characterization. Since this is not the case, I would use the term sequences rather than strains.
Line 75: Please modify: G and C nucleotides were highly enriched in PPV genomes.
Line 85: Which species the authors refer to?
Line 111: Please correct …”one extra VEGF gene at the left end..”.
Line 120: As quoted above, there are more PPVs circulating in the world during the past 20 years than just the studied here. Please adjust.
Line 126: Is this increased prevalence favored by availability of sequencing techniques. Are the Orf outbreaks of compulsory declaration in China and India? Many areas in which Orf is endemic have not the obligation to declare outbreaks, therefore conclusions should be restricted to the studied area.
Line 132: Why this was conducted with 5 different segments and not the complete genomes available at GenBank? As authors claim in the dN/dS ratio analysis, big differences may exist among different genomic segments.
Line 170: These methodologic details are appropriate for the Material and Methods section.
Line 183 and elsewhere: Was positive selection only found in VIR and GIF genes?
Line 208: Do you mean For instance,..?
Line 210-211: Sentence not clear. Significant differences among PPVs in GC content but identical nucleotide bias? Please clarify. Is this supporting previous knowledge on the mechanisms of such GC content? Please specify if so.
Line 212: What is the author’s opinion about considering the different PPVs as different genotypes of the same etiological agent?
Lines 216-218: This is not consistent with the description provided in the Introduction. Is ANK a structural protein? Is VEGF involved in virus assembly? Etc.
Line 224: More details on the role of IL10 in immunity and PPV infection should be included.
Line 228: Suggestion to modify: The proportion of ORFV strains SEQUENCED and AVAILABLE indicate a sharply increase in the past 20 years.
Lines 234-237: Please rephrase. Could be the close phylogenetic clustering a consequence of reduced diversity?
Line 243: Why these mutation rates are compared with such divergent viruses? How would be the comparison with closer counterparts, such as poxviruses?
Line 244: Could be the long-lasting interaction with the immune response the cause of the positive selection observed in VIR gene, rather than the explanation offered. If VIR were not so important for replication, why should it be maintained in all PPVs across the evolution?
Line 248: Same comment as for VIR. GIF is likely interacting with proteins of the immune system.
Line 252: Substitute “higher” by “high” or indicate the two categories compared.
Lines 251-258: This paragraph should be improved for English editing.
Line 278: Is this temporal interval correct?
Lines 290-297: This part is in capital character.
Table 2: Why is B2L in bold character?
Figure 3: Please indicate the reference sequence and include nucleotide position ruler. Please define ML in the caption.
Figure 6: This figure is not clear to me. While a decreased diversity is claimed in the manuscript, it does not seem the case for GIF or VIR for which the blue color is predominant in recent years.
Author Response
To Reviewer1
We are very grateful for your assessment of our work. Thank you very much for your time and effort in reviewing our manuscript. We greatly appreciate your useful comments. Please find our responses to each of your suggestions in the following section.
Reviewer1:
Comments and Suggestions for Authors
Yao et al. present here an interesting work in which phylogeny, genome structure and evolution features of Parapoxviruses using previously deposited sequences from 2000 to 2020 are depicted. Although the manuscript contains important and interesting knowledge, some important drawbacks impede its publication in the present form.
Major points:
1.- The manuscript needs deep English editing and usage revision. Some parts of the manuscript are really difficult to follow. Some examples as specific comments below.
Response: Thank you very much for reviewing our manuscript. Our paper was edited by an English expert. He changed and corrected so many sentences in the current manuscript.
2.- The manuscript lacks of accuracy and inconsistences in some parts. For example, PPV infection is claimed to cause high mortality rates.
Response: Thank you very much for your useful suggestions. We have corrected this part in our revised manuscript: “ORFV, the prototype species of PPVs can cause lesions in lips and breast of suckling and grazing animals with high morbidity and low mortality rates”. (line 61-62)
3.- In general conclusions are overstated, i.e., the authors claim for an increased ORF prevalence without taking into account the availability of sequences. The possibility that countries suffering from Orf virus infection without reporting cases or uploading sequences to GenBank cannot be ruled out. Thus, conclusions should be softened. Parapoxvirus infections of ruminants and handlers are endemic worldwide. For example Lederman et al 2007. Clin Infect Dis. Taking into account that most sequences analyzed were from China and India, conclusions and objectives should be dimensioned.
Response: We appreciate the reviewer’s great suggestion. In our research, we just took into account all available sequences in GenBank database, as your opinion, we have softened our conclusion in the current manuscript.
4.- Phylogenetic evolution is based on 5 different genetic segments. Since the different genes exhibited different genetic features, such as different dN/ds ratio, why the authors think that evolution inferences would be that accurate? Complete genome analysis would offer a more precise picture.
Response: Thank you very much for reviewing our manuscript. Yes, complete genome analysis would offer a more precise picture, but actually, the complete genome sequences in GenBank database are not enough to study. Additionally, the segmented genome feature of parapoxvirus allows reassortment of segments from different viruses, generating novel parapoxvirus with pandemic potential. Therefore, we investigated the roles of the evolution of parapoxvirus in the seven viral gene sequences.
Specific points:
Line 23: ORFV please define.
Response: Thank you for your suggestions. We have added the definition of ORFV in our revised manuscript. (line 24)
Line 24: This work is not unveiling mechanisms that determine the high GC content found in PPVs, since reasons could be variable, different mechanisms for the different viruses cannot be ruled out.
Response: Thank you for your suggestions. According to your opinion, we have removed this conclusion in the current manuscript.
Line 28: Which outbreak the authors refer to? Please specify.
Response: Thank you for your suggestions. On the basis of all available sequences, it contains all outbreaks during 2010-2018. (line 29-30)
Lines 38-39: PPVs are also present in domestic ruminants. Please correct: “non-systemic skin lesions in wild and domestic animals”
Response: Thank you for your suggestions. We have corrected this sentence: “Parapoxvirus (PPV) is a chordopoxvirus that causes non-systemic skin lesions in wild and domestic animals.”. (line 43-44)
Lines 47-48: The sentence is not clear. Please re-phrase.
Response: Thank you for your suggestions. We have revised this sentence in the current manuscript. (line 54-55)
Line 49: Better “Humans can be infected…”
Response: Thank you for your suggestions. We have changed this sentence in our revised manuscript. (line 56)
Line 51: South America is repeated twice.
Response: We apologize for our typing mistake. We polished our current manuscript and changed all the capitalization mistakes.
Line 54: ORFV infection is characterized by high morbility and low mortality rates. Please correct.
Response: Thank you for your suggestions. We have corrected this sentence in our revised manuscript. (line 61-62)
Lines 59-60: The last two sentences of the paragraph are confusing. Please rephrase.
Response: Thank you for your suggestions. We have corrected this sentence in our revised manuscript. (line 66-69)
Lines 62-67: The whole paragraph is awkward and confusing. No relation with vaccination is presented in the manuscript, and the selection pressure due to vaccination is not clearly explained.
Response: Thank you for your suggestions. We have revised this paragraph in the current manuscript. (line 70-81)
Line 68: This seems to be overstated. Since PPVs are endemic in USA or Mediterranean countries such as Italy or Spain, and no data are analyzed in this work, I would suggest to soften some global conclusions.
Response: Thank you for your suggestions. The conclusion has been softened in the revised manuscript. (line 82)
Line 70: Investigating genomic features of different strains would imply isolation of the corresponding viruses and genetic characterization. Since this is not the case, I would use the term sequences rather than strains.
Response: Thank you for your suggestions. We have used the term sequences instead of strains in our revised manuscript. (line 85)
Line 75: Please modify: G and C nucleotides were highly enriched in PPV genomes.
Response: Thank you for your suggestions. We have modified this sentence in the revised manuscript. (line 92)
Line 85: Which species the authors refer to?
Response: Thank you for your suggestions. We have added the explanation of these species in the revised manuscript. (line 103-104)
Line 111: Please correct …”one extra VEGF gene at the left end..”.
Response: Thank you for your suggestions. We have corrected this sentence in the revised manuscript. (line 122)
Line 120: As quoted above, there are more PPVs circulating in the world during the past 20 years than just the studied here. Please adjust.
Response: Thank you for your suggestions. We have adjusted this sentence into: “266 available complete sequences of PPV B2L genes were retrieved from GenBank in this study”. (line 127)
Line 126: Is this increased prevalence favored by availability of sequencing techniques. Are the Orf outbreaks of compulsory declaration in China and India? Many areas in which Orf is endemic have not the obligation to declare outbreaks, therefore conclusions should be restricted to the studied area.
Response: Thank you for your suggestions. We have adjusted the conclusion in the revised manuscript. (line 134-135)
Line 132: Why this was conducted with 5 different segments and not the complete genomes available at GenBank? As authors claim in the dN/dS ratio analysis, big differences may exist among different genomic segments.
Response: Thank you for your suggestions. In our research, we constructed phylogenetic trees for all of the seven segments. Because there are only 17 complete genome sequences available at GenBank, they may not enough be analyzed in our study.
Line 170: These methodologic details are appropriate for the Material and Methods section.
Response: Thank you for your suggestions. We have put these methodologic details into the Material and Methods section. (line 167-168; line 326-327)
Line 183 and elsewhere: Was positive selection only found in VIR and GIF genes?
Response: No, positive selection pressure occurred at each gene.
Line 208: Do you mean For instance,..?
Response: We apologize for our typing mistake. We polished our current manuscript and changed all the capitalization mistakes.
Line 210-211: Sentence not clear. Significant differences among PPVs in GC content but identical nucleotide bias? Please clarify. Is this supporting previous knowledge on the mechanisms of such GC content? Please specify if so.
Response: Thank you very much for reviewing our manuscript. We have revised the discussion of this part and removed this sentence in the current manuscript.
Line 212: What is the author’s opinion about considering the different PPVs as different genotypes of the same etiological agent?
Response: Thank you very much for your useful suggestions. We have added the discussion of this part in the revised manuscript. (line 249-250)
Lines 216-218: This is not consistent with the description provided in the Introduction. Is ANK a structural protein? Is VEGF involved in virus assembly? Etc.
Response: Thank you very much for your useful suggestions. We have revised the discussion and polished our current manuscripts.
Line 224: More details on the role of IL10 in immunity and PPV infection should be included.
Response: We appreciate the reviewer’s great suggestion. We have added more explanation in discussion and please check it. (line 259-263)
Line 228: Suggestion to modify: The proportion of ORFV strains SEQUENCED and AVAILABLE indicate a sharply increase in the past 20 years.
Response: Thank you for your suggestion. We have modified this sentence in our revised manuscript. (line 267-269)
Lines 234-237: Please rephrase. Could be the close phylogenetic clustering a consequence of reduced diversity?
Response: Thank you for your suggestion. We have added more explanation and revised this sentence in the current manuscript. (line 275-278)
Line 243: Why these mutation rates are compared with such divergent viruses? How would be the comparison with closer counterparts, such as poxviruses?
Response: Thank you for your insightful suggestion. We have modified this sentence in our revised manuscript. (line 287)
Line 244: Could be the long-lasting interaction with the immune response the cause of the positive selection observed in VIR gene, rather than the explanation offered. If VIR were not so important for replication, why should it be maintained in all PPVs across the evolution?
Response: Thank you for your suggestion. We have modified this sentence in the revised manuscript. (line 285-287)
Line 248: Same comment as for VIR. GIF is likely interacting with proteins of the immune system.
Response: Thank you for your suggestion. According to your comment, we have revised this sentence in the current manuscript. (line 293-294)
Line 252: Substitute “higher” by “high” or indicate the two categories compared.
Response: Thank you for your suggestion. We have modified this sentence in the current manuscript.
Lines 251-258: This paragraph should be improved for English editing.
Response: Thank you for your suggestion. We have revised this paragraph to improve the English editing. (line 293-299)
Line 278: Is this temporal interval correct?
Response: Yes, this is correct, because we downloaded all B2L sequences from NCBI nucleotide database, and the collection date of this B2L dataset is during 1974 – 2018.
Lines 290-297: This part is in capital character.
Response: Thank you for your suggestion. We have corrected our current manuscripts. (line 338-347)
Table 2: Why is B2L in bold character?
Response: Thank you for your suggestion. We have polished our current manuscripts.
Figure 3: Please indicate the reference sequence and include nucleotide position ruler. Please define ML in the caption.
Response: Thank you for your suggestion. We have indicated the reference sequence and nucleotide position ruler in figure 3, and defined ML in the caption.
Figure 6: This figure is not clear to me. While a decreased diversity is claimed in the manuscript, it does not seem the case for GIF or VIR for which the blue color is predominant in recent years.
Response: Thank you for your insightful suggestion. We have revised the conclusion about figure 6 in order to improve the quality of the manuscript. (line 151)
Reviewer 2 Report
Yao et all present an in silico analysis of PPV genomes and several specific genes in greater depth to tease out evolutionary selection pressures during a ~20 yr period. While the analyses were well found I believe one major concern needs to be addressed. The minor concerns have mostly to do with English usage and I have detailed those that I noticed. A thorough copyediting is in order though the text is understandable.
Major:
The authors spend a great deal of time discussing GC content but no discussion of the ramifications of GC content on the sequencing. As the authors did not perform the sequencing themselves and took the information from an uncurated source I am concerned that the known issues with high GC amplification could have fixed erroneous base calls in the sequencing in an unpatterned and uneven manner. The same sample sequenced multiple times can provide greatly divergent sequencing depth. Sequence depth can be incredibly important to consensus sequencing. To address this I recommend cleaning the sequence of non-segregating sites and rerunning the mutation rate analyses. Additionally pulling assemblies that provided the sequences used if available from SRA and looking at the intrahost population could be utilized to support confidence in the base calls.
Minor:
19 … However, it is still not fully understood the viral
Awkward rephrase
20 prevalence and evolution of PPV coding sequences. Here, we performed a comparative approach 21 integrating viral genetics, molecular selection pressure and genomic structure to investigate the
Add caveat ‘in silico’ molecular selection pressure not directly assessed.
47 PPVs have a global distribution, and the ruminant flocks are disturbed while the virus emerged
48 in many continents [7-10]. This virus is known to infect mammals such as: goats, sheep, cattle, camels
49 reindeer and other ruminants [7, 11]. Human can also be infected by PPVs, following direct exposure
50 to skin lesions of infected mammals.
51 PPVs infections are in place in a lot of regions of the world including
52 South America [7, 12-16]. Currently
59 … Except for the most 60 important viruses (ORFV and BPSV), PCPV and PVNZ both maintained in mammals.
Rewrite. Unclear statement.
64 … And they usually have a high mutation rate with
Unclear antecedent.
66 … The PPV has a large genome, comprising nearly 134~139 kb in size. Consequently, 67 mutation is more likely to occur in PPVs under vaccination and selection pressures [27, 28].
Unclear causal link. Explain.
81 likely to occur
196 PPV infections normally occur in animals and even in humans throughout the world [7, 13, 15, 197 16, 30, 31].
Awkward. Rephrase.
197 16, 30, 31]. However, comprehensive studies on PPVs are still lacking.
208 strategies, but may show different GC frequencies [36]. For instantance
283 … drown …
Drawn
321 … Besides, the genetic characteristics
Awkward rephrase
Author Response
To Reviewer2
We are very grateful for your assessment of our work. Thank you very much for your time and effort in reviewing our manuscript. We greatly appreciate your useful comments. Please find our responses to each of your suggestions in the following section.
Reviewer 2:
Comments and Suggestions for Authors
Yao et all present an in silico analysis of PPV genomes and several specific genes in greater depth to tease out evolutionary selection pressures during a ~20 yr period. While the analyses were well found I believe one major concern needs to be addressed. The minor concerns have mostly to do with English usage and I have detailed those that I noticed. A thorough copyediting is in order though the text is understandable.
Major:
The authors spend a great deal of time discussing GC content but no discussion of the ramifications of GC content on the sequencing. As the authors did not perform the sequencing themselves and took the information from an uncurated source I am concerned that the known issues with high GC amplification could have fixed erroneous base calls in the sequencing in an unpatterned and uneven manner. The same sample sequenced multiple times can provide greatly divergent sequencing depth. Sequence depth can be incredibly important to consensus sequencing. To address this I recommend cleaning the sequence of non-segregating sites and rerunning the mutation rate analyses. Additionally pulling assemblies that provided the sequences used if available from SRA and looking at the intra host population could be utilized to support confidence in the base calls.
Response: We appreciate the reviewer’s great suggestion. We downloaded all PPV sequences from GenBank database, and filtered the available sequences to do further analysis. Therefore, each sequence quality is credibly in our study.
Minor:
19 … However, it is still not fully understood the viral
Awkward rephrase
Response: Thank you very much for reviewing our manuscript. We have revised this sentence in the current manuscript. (line 21-22)
20 prevalence and evolution of PPV coding sequences. Here, we performed a comparative approach 21 integrating viral genetics, molecular selection pressure and genomic structure to investigate the
Add caveat ‘in silico’ molecular selection pressure not directly assessed.
Response: Thank you very much for your useful suggestions. We have revised this sentence in the current manuscript. (line 22-23)
47 PPVs have a global distribution, and the ruminant flocks are disturbed while the virus emerged
48 in many continents [7-10]. This virus is known to infect mammals such as: goats, sheep, cattle, camels
49 reindeer and other ruminants [7, 11]. Human can also be infected by PPVs, following direct exposure
50 to skin lesions of infected mammals.
51 PPVs infections are in place in a lot of regions of the world including
52 South America [7, 12-16]. Currently
59 … Except for the most 60 important viruses (ORFV and BPSV), PCPV and PVNZ both maintained in mammals.
Rewrite. Unclear statement.
Response: Thank you for your suggestion. We have revised these sentences in the current manuscript. (line 54-60)
64 … And they usually have a high mutation rate with
Unclear antecedent.
Response: Thank you for your suggestion. We have revised these sentences in the current manuscript. (line 70-81)
66 … The PPV has a large genome, comprising nearly 134~139 kb in size. Consequently, 67 mutation is more likely to occur in PPVs under vaccination and selection pressures [27, 28].
Unclear causal link. Explain.
Response: Thank you for your suggestion. We have revised these sentences in the current manuscript. (line 75-78)
81 likely to occur
196 PPV infections normally occur in animals and even in humans throughout the world [7, 13, 15, 197 16, 30, 31].
Awkward. Rephrase.
Response: Thank you for your suggestions. We polished our current manuscripts and please check it. (line 98-99; line 235)
197 16, 30, 31]. However, comprehensive studies on PPVs are still lacking.
208 strategies, but may show different GC frequencies [36]. For instantance
283 … drown …
Drawn
321 … Besides, the genetic characteristics
Awkward rephrase
Response: We apologize for our typing mistake. We polished our current manuscript and changed all the capitalization mistakes. (line 236; line 332; line 375-376)
Reviewer 3 Report
Review of Genomic features and evaluation of the Parapoxvirus during the past two decades.
Yao et al present the genomic characteristics and an analysis of divergence within the Parapoxvirus family over the last twenty years. Overall, the experiments are performed well, following commonly accepted methods. The paper should be reviewed for English use because substantial changes are necessary to improve readability.
Specific comments below
P2 line 43, what is "evenly"
P2 paragraph beginning on line 47, 1st sentence rephrase for clarity. What is a ruminant flock, what is meant by disturbed? Are ruminants the only hosts of the parapoxviruses? What is the course of human disease, severe, mild, systemic, limited to exposure site?
P2 line 59, rephrase "significant immunity" do the authors mean a memory response? If the disease is not severe and clears, then significant immunity has limited the spread and severity of disease within that subject.
P2 line 60. The public health threat needs to be expanded upon, how many cases per year, how many cases have increased over the last 10, 15, or 20 yrs? What is the data to support parapoxviruses as public health threats?
P2 line 63, statement "viruses may acquire greater virulence through genetic variation and recombination, as a result of the rapid transmission in vaccinated populations", please provide an explanation and example, not just the references. How does the wild type virus transmit more rapidly in a vaccinated population? Doesn't that counter the logic of vaccination and herd immunity?
Time to most common recent ancestor analysis?
P6, line 147, please clarify "despite their prevalence in the world for the last 20 years", what is meant?
P8 line 171. Are codon-based phylogenetic models suitable for hosts without complete genomes and transcriptome profiles?
P9 line 204, what's the connection between GC composition of the PPVs and the other viruses mentioned? What about compared to orthopoxviruses?
Beginning on line 228, what is meant by the “proportion of ORFV strains have undergone a sharply increase”. Are there more strains, if so, how is a strain defined vs. an isolate? What is accepted by ICTV or other standard for the PPVs in regards to sequence identity for a strain vs. an isolate? What are the selection pressures driving ORFV expansion?
Author Response
To Reviewer3
We are very grateful for your assessment of our work. Thank you very much for your time and effort in reviewing our manuscript. We greatly appreciate your useful comments. Please find our responses to each of your suggestions in the following section.
Reviewer 3:
Comments and Suggestions for Authors
Review of Genomic features and evaluation of the Parapoxvirus during the past two decades.
Yao et al present the genomic characteristics and an analysis of divergence within the Parapoxvirus family over the last twenty years. Overall, the experiments are performed well, following commonly accepted methods. The paper should be reviewed for English use because substantial changes are necessary to improve readability.
Specific comments below
P2 line 43, what is "evenly"
Response: Thank you very much for reviewing our manuscript. We have revised this sentence the current manuscript. (line 48-49)
P2 paragraph beginning on line 47, 1st sentence rephrase for clarity. What is a ruminant flock, what is meant by disturbed? Are ruminants the only hosts of the parapoxviruses? What is the course of human disease, severe, mild, systemic, limited to exposure site?
Response: Thank you for your suggestion. We have added the explanation and revised this paragraph in the current manuscript. (line 54-60)
P2 line 59, rephrase "significant immunity" do the authors mean a memory response? If the disease is not severe and clears, then significant immunity has limited the spread and severity of disease within that subject.
Response: Thank you very much for your useful suggestions. We have modified this sentence in the current manuscript. (line 67)
P2 line 60. The public health threat needs to be expanded upon, how many cases per year, how many cases have increased over the last 10, 15, or 20 yrs? What is the data to support parapoxviruses as public health threats?
Response: Thank you for your suggestion. We have modified this sentence in the revised manuscript, please check it. (line 68-69)
P2 line 63, statement "viruses may acquire greater virulence through genetic variation and recombination, as a result of the rapid transmission in vaccinated populations", please provide an explanation and example, not just the references. How does the wild type virus transmit more rapidly in a vaccinated population? Doesn't that counter the logic of vaccination and herd immunity?
Time to most common recent ancestor analysis?
Response: Thank you very much for reviewing our manuscript. We have added more explanation about this statement, please check it. (line 73-78)
P6, line 147, please clarify "despite their prevalence in the world for the last 20 years", what is meant?
Response: We appreciate the reviewer’s great suggestion. We have modified this sentence in the revised manuscript. (line 153-155)
P8 line 171. Are codon-based phylogenetic models suitable for hosts without complete genomes and transcriptome profiles?
Response: Thank you for your suggestion. Yes, and we used the complete sequences of virus, it is suitable for our data. Additionally, we have put these methodologic details into the Material and Methods section. (line 326-327)
P9 line 204, what's the connection between GC composition of the PPVs and the other viruses mentioned? What about compared to orthopoxviruses?
Response: Thank you for your suggestion. We just removed this sentence because we cannot find out the references about orthopoxviruses GC composition.
Beginning on line 228, what is meant by the “proportion of ORFV strains have undergone a sharply increase”. Are there more strains, if so, how is a strain defined vs. an isolate? What is accepted by ICTV or other standard for the PPVs in regards to sequence identity for a strain vs. an isolate? What are the selection pressures driving ORFV expansion?
Response: Thank you very much for your useful suggestions. We have revised this sentence in the current manuscript. (line 267-269)
Round 2
Reviewer 1 Report
Authors have successfully assessed the changes suggested by reviewers, however, there are still abundant grammatical errors and confusing sentences. A detailed report including some of them is provided.
In general authors should carefully revise typing and English editing. In addition, grammatical revision by an expert is highly recommended as it will significantly improve the manuscript.
Line 23: Is the whole genome segmented that is analysed? If not, please rephrase as for example: “…phylogenetic diversity of X different genes of PPV”
Line 24: Authors refer here to an outbreak that has not been presented before. Where was described? Which species were affected?
Line 26: Substitution rates were between x and y.
Line 27: PPV VIR gene evolved. Delete was.
Line 27-29: Which protein-coding regions? This sentence is a bit confusing since these genes may refer to VIR (previous sentence).
Line 30: Which is the new perspective? Revise sentence: “…could help to create…”
Line 37: Connecting its terminal is confusing.
Line 41: Perhaps vital is not the right word when referring to viruses.
Line 47: What authors mean with plenty?
Line 55: Are really PCPV and PVNZ affecting public health? Perhaps animal health could be more appropriate.
Lines 56-65: Despite authors have modified this paragraph by adding information about other poxviruses, it is still unclear the relationship between vaccination and the selection pressure imposed to the virus, that authors aim to explain. Examples offered are not derived from vaccination campaigns or at least not specified. In any case, vaccination campaigns are not detailed in the different countries in terms of inocula, vaccination protocol, etc. that may allow authors to speculate about it.
Line 102: Awkward expression. Please rephrase. Suggestion “…contains large genes encoding functional proteins…”
Line 107: What the authors refer to with “This”? The presence of accessory proteins in PPVs suugested that the right end of BPSV genomes were similar? Please re-phrase.
Line 112: Please correct: ANK genes that may be involved in host range.”
Line 120: Which change? Is that a specific change? Maybe “changes” is a more adequate word in this case?
Line 120: As highlighted in the previous report references to the strains circulating in the world is not correct with the data analysed here since only previously deposited sequences were considered.
Line 124: This idea is just mentioned in the previous sentence.
Line 145: Do you mean starting mainly in 2010?
Line 166: What does mean that Bayesian calculated the mean evolutionary rates? Do you mean Bayesian analyses?
Line 169: “…with a substitution rate of…”
Line 207: This sentence is not clear. Do you mean that divergence among PPVs was demonstrated by analyzing complete sequences?
Line 209: Since PPVs englobe more than one virus should be considered plural.
Line 210: “It has been suggested…”
Line 217: Please rephrase, specially the part saying “…particular macrophage role and Th1 effector role.”
Line 218: “…these divergent genomic features…”
Line 227-229: Please rephrase.
Line 231-232. The last sentence in this paragraph is not really informative.
Line 238: Authors should explain what is intended with variability in viral synthesis and replication process; and how is it reflected by a high substitution rate? Could it have been observed in a gene not involved in the replication process?
Line 241: Substitute less by lower. For the GIF and VIR genes?
Line 242: This demonstrates that purifying selection is the major force driving selection in the segments analysed, may be not in the whole PPVs’ genome. Please adjust.
Line 244: Which proteins are likely in contact with immune system proteins? Do authors have any reference to support this?
Line 246: The fact that VTLF-1 shows purifying selection is not a speculation, is derived from the evolutionary rate calculated.
Lines 248-250: Authors should provide an explanation for their findings in the viral infection context, ideally in the PPV context.
Line 255: Revise character size.
Line 257: “…each PPV isolate was determined…”
Line 260-261: Suggestion: Partial sequences from the aforementioned genes were obtained from the complete genomes available at GenBank.
Line 266: Typing error.
Line 313: Animal or virus species?
Lines 313-322: The whole section should be re-written.
Table 2: Why is B2L underlined? The title should be gene rather than protein since comparisons are made with nucleotides.
Author Response
Comments and Suggestions for Authors
Authors have successfully assessed the changes suggested by reviewers, however, there are still abundant grammatical errors and confusing sentences. A detailed report including some of them is provided.
In general authors should carefully revise typing and English editing. In addition, grammatical revision by an expert is highly recommended as it will significantly improve the manuscript.
Response: Thank you very much for reviewing our manuscript. The manuscript was revised by an English expert. He changed and corrected so many sentences in the current manuscript.
Line 23: Is the whole genome segmented that is analysed? If not, please rephrase as for example: “…phylogenetic diversity of X different genes of PPV”
Response: Thank you very much for your useful suggestions. We have corrected this sentence to: “we estimated the phylogenetic diversity of seven different genes of PPV”. (line 23)
Line 24: Authors refer here to an outbreak that has not been presented before. Where was described? Which species were affected?
Response: We appreciate the reviewer’s great suggestion. We have revised the manuscript to make it more clear. And ORFV was mainly affected goat. (line 24-25)
Line 26: Substitution rates were between x and y.
Response: Thank you very much for reviewing our manuscript. We have revised this sentence to “substitution rates ranged from 3.56×10-5 to 4.21×10-4”. (line 26)
Line 27: PPV VIR gene evolved. Delete was.
Response: Thank you for your suggestions. We have deleted “was” in the revised manuscript. (line 27)
Line 27-29: Which protein-coding regions? This sentence is a bit confusing since these genes may refer to VIR (previous sentence).
Response: Thank you for your suggestions. According to your opinion, this sentence has been revised to “In these seven protein-coding regions”. (line 27)
Line 30: Which is the new perspective? Revise sentence: “…could help to create…”
Response: Thank you for your suggestions. It has been corrected in the revised manuscript. And the new perspective means to create prevention and control strategies for virus. (line 30)
Line 37: Connecting its terminal is confusing.
Response: Thank you for your suggestions. This sentence has been revised to: “The PPV genome is approximately 134~139 kb in size with an extremely high GC content, which goes up to 65%”. (line 38-39)
Line 41: Perhaps vital is not the right word when referring to viruses.
Response: Thank you for your suggestions. We have corrected this word to “important”. (line 41)
Line 47: What authors mean with plenty?
Response: Thank you for your suggestions. We have corrected “plenty” to “many”. (line 47)
Line 55: Are really PCPV and PVNZ affecting public health? Perhaps animal health could be more appropriate.
Response: Thank you for your suggestions. We have corrected “public health” to “animal health” in the revised manuscript. (line 47)
Lines 56-65: Despite authors have modified this paragraph by adding information about other poxviruses, it is still unclear the relationship between vaccination and the selection pressure imposed to the virus, that authors aim to explain. Examples offered are not derived from vaccination campaigns or at least not specified. In any case, vaccination campaigns are not detailed in the different countries in terms of inocula, vaccination protocol, etc. that may allow authors to speculate about it.
Response: Thank you for your suggestions. We have added more explanation about the relationship between vaccination and the selection pressure imposed to the virus. (line 61-64)
Line 102: Awkward expression. Please rephrase. Suggestion “…contains large genes encoding functional proteins…”
Response: Thank you for your suggestions. We have revised this sentence to: “a large number of protein-coding regions which were encoding functional proteins”. (line 102)
Line 107: What the authors refer to with “This”? The presence of accessory proteins in PPVs suugested that the right end of BPSV genomes were similar? Please re-phrase.
Response: Thank you for your suggestions. This sentence has been revised to: “These results suggested that compared with other PPVs, VRGF genes were existing in the left terminal region of BPSV genomes.” (line 108-109)
Line 112: Please correct: ANK genes that may be involved in host range.”
Response: Thank you for your suggestions. This sentence has been revised in the current manuscript. (line 112)
Line 120: Which change? Is that a specific change? Maybe “changes” is a more adequate word in this case?
Response: We apologize for our typing mistake. We have corrected “change” to “changes”. (line 120)
Line 120: As highlighted in the previous report references to the strains circulating in the world is not correct with the data analysed here since only previously deposited sequences were considered.
Response: Thank you for your suggestions. We have corrected this sentence in the current manuscript. (line 122)
Line 124: This idea is just mentioned in the previous sentence.
Response: Thank you for your suggestions. We have removed this sentence in the revised manuscript.
Line 145: Do you mean starting mainly in 2010?
Response: Thank you for your suggestions. Yes, it suggested that a decreased diversity was arising since 2010.
Line 166: What does mean that Bayesian calculated the mean evolutionary rates? Do you mean Bayesian analyses?
Response: Thank you for your suggestions. This sentence has been revised to: “Bayesian coalescent approach was performed to calculate…” (line 165-166)
Line 169: “…with a substitution rate of…”
Response: Thank you for your suggestions. We have revised this sentence to: “…with a substitution rate of…” (line 168)
Line 207: This sentence is not clear. Do you mean that divergence among PPVs was demonstrated by analyzing complete sequences?
Response: Thank you for your suggestions. Yes, it means that divergence among PPVs was demonstrated by analyzing complete sequences.
Line 209: Since PPVs englobe more than one virus should be considered plural.
Response: Thank you for your suggestions. We have revised this sentence in the current manuscript. (line 211)
Line 210: “It has been suggested…”
Response: Thank you for your suggestions. We have revised it to: “It has been suggested…” (line 209)
Line 217: Please rephrase, specially the part saying “…particular macrophage role and Th1 effector role.”
Response: Thank you for your suggestions. It has been revised in the current manuscript. (line 219-220)
Line 218: “…these divergent genomic features…”
Response: Thank you for your suggestions. We have revised the “feature” to “features”. (line 218)
Line 227-229: Please rephrase.
Response: Thank you for your suggestions. We have revised this sentence to: “It has been indicated that under the selective pressure, ORFV eventually became the dominant species during the 2010 – 2018.” (line 229-230)
Line 231-232. The last sentence in this paragraph is not really informative.
Response: Thank you for your suggestions. We have removed this sentence in the revised manuscript.
Line 238: Authors should explain what is intended with variability in viral synthesis and replication process; and how is it reflected by a high substitution rate? Could it have been observed in a gene not involved in the replication process?
Response: Thank you very much for your useful suggestions. We feel sorry about it because the relative references are lack to explain this process.
Line 241: Substitute less by lower. For the GIF and VIR genes?
Response: Thank you very much for your useful suggestions. We have revised this sentence in the current manuscript. (line 240)
Line 242: This demonstrates that purifying selection is the major force driving selection in the segments analysed, may be not in the whole PPVs’ genome. Please adjust.
Response: Thank you very much for your useful suggestions. This sentence has been revised to: “the overall rates of dN/dS were mostly less than 1, demonstrating that purifying selection was the major force to drive the evolution of PPV genes.” (line 239-241)
Line 244: Which proteins are likely in contact with immune system proteins? Do authors have any reference to support this?
Response: Thank you very much for your useful suggestions. We have added more reference to support it. (line 244)
Line 246: The fact that VTLF-1 shows purifying selection is not a speculation, is derived from the evolutionary rate calculated.
Response: We appreciate the reviewer’s great suggestion. We have revised this sentence to: “Our results suggested that…” (line 245)
Lines 248-250: Authors should provide an explanation for their findings in the viral infection context, ideally in the PPV context.
Response: Thank you for your suggestions. We have added more explanation about this part in the revised manuscript. (line 249-253)
Line 255: Revise character size.
Response: We appreciate the reviewer’s great suggestion. We have revised character size in the current manuscript. (line 253)
Line 257: “…each PPV isolate was determined…”
Response: Thank you very much for reviewing our manuscript. We have corrected “were” to “was” in the revised manuscript. (line 255)
Line 260-261: Suggestion: Partial sequences from the aforementioned genes were obtained from the complete genomes available at GenBank.
Response: Thank you very much for reviewing our manuscript. We have revised this sentence in the current manuscript. (line 259-260)
Line 266: Typing error.
Response: We apologize for our typing mistake. We have corrected the typing error in the revised manuscript. (line 268)
Line 313: Animal or virus species?
Response: Thank you very much for reviewing our manuscript. We have revised this phrase to “virus species”. (line 311)
Lines 313-322: The whole section should be re-written.
Response: Thank you for your suggestions. We have revised this section in the current manuscript. (line 315-322)
Table 2: Why is B2L underlined? The title should be gene rather than protein since comparisons are made with nucleotides.
Response: Thank you very much for reviewing our manuscript. We have corrected table 2 in the revised manuscript.
Reviewer 2 Report
The authors have failed to address a major problem with their research study. Having downloaded and 'filtered' according to some undescribed quality parameters is not a sufficient rebuttal.
Nowhere in the text do you describe either how high GC content can affect PCR and thereby next gen sequencing nor why it is of importance to from a biological perspective.
The methods describe filtering as restricting to those with sufficient metadata (host, collection date, sampling location) and no sequence quality filtering is described.
Missing metadata of importance are sequencing protocol and platform, protocol deviations for sequencing high GC content loci, etc. How diverse is the sequencing source (lab responsible for generating the data) for each gene? If for instance the preponderance of gene sequence is form a minority of labs and another from a different minority of labs then the protocols and sequencing platform could heavily influence the aggregation of errors in the consensus sequence and present an alternate explanation for their mutation rate findings.
If you do not sufficiently understand the problem I am describing please ask for more information.
Minor issues were resolved sufficiently.
Author Response
Comments and Suggestions for Authors
The authors have failed to address a major problem with their research study. Having downloaded and 'filtered' according to some undescribed quality parameters is not a sufficient rebuttal.
Nowhere in the text do you describe either how high GC content can affect PCR and thereby next gen sequencing nor why it is of importance to from a biological perspective.
The methods describe filtering as restricting to those with sufficient metadata (host, collection date, sampling location) and no sequence quality filtering is described.
Missing metadata of importance are sequencing protocol and platform, protocol deviations for sequencing high GC content loci, etc. How diverse is the sequencing source (lab responsible for generating the data) for each gene? If for instance the preponderance of gene sequence is form a minority of labs and another from a different minority of labs then the protocols and sequencing platform could heavily influence the aggregation of errors in the consensus sequence and present an alternate explanation for their mutation rate findings.
If you do not sufficiently understand the problem I am describing please ask for more information.
Minor issues were resolved sufficiently.
Response: Thank you for your suggestion. In the section of Materials and Methods, we downloaded the sequences in NCBI GenBank database, what we did is similar with the prior bioinformatics research, we followed the way of published reference. For example, in 2019, researchers downloaded the 217 complete coding sequences of Influenza C Virus to explore their genetic evolution and molecular Selection (Zhang et al., 2019). All sequences we selected were from the GenBank database and have been published.